# Using Remote Sensing to Identify Urban Fringe Areas and Their Spatial Pattern of Educational Resources: A Case Study of the Chengdu-Chongqing Economic Circle

**Wei Lu** [1,2], **Yuechen Li** [1,2,*], **Rongkun Zhao** [1,2] **and Yue Wang** [1,2]

1 Chongqing Jinfo Mountain National Field Scientific Observation and Research Station for Karst Ecosystem, Chongqing Engineering Research Center for Remote Sensing Big Data Application, School of Geographical Sciences, Southwest University, Chongqing 400715, China; my19981226@email.swu.edu.cn (W.L.); zrk1998@email.swu.edu.cn (R.Z.); wwyue1998@email.swu.edu.cn (Y.W.)
2 Key Laboratory of Monitoring, Evaluation and Early Warning of Territorial Spatial Planning Implementation, Ministry of Natural Resources, Chongqing 401147, China
* Correspondence: liyuechen@swu.edu.cn; Tel.: +86-23-6825-3912

**Abstract:** Rapid urbanization has already caused many impacts, such as environmental degradation and imbalanced resource allocation. As the frontiers of urbanization, urban fringe areas (UFAs) present both urban and rural characteristics and undergo complex socio-economic structural changes. Accurately identifying the spatial extent of UFAs is highly significant because it contributes to understanding the pattern of urban spatial expansion and guides future urban planning. However, existing methods are strongly affected by subjective factors. To solve this problem, this study presents a new approach to identifying UFAs, with the Chengdu-Chongqing economic circle as the study area. The new method achieved an identification accuracy of 74.2%, effectively eliminated some noise points, and reduced the influence of subjective factors. From an applied perspective, this study employed the Geo-information Tupu and density-field-based hotspot detector to analyze the spatial pattern of educational resources. Overall, the results showed that hotspots of educational resources are concentrated in places with good transportation or near urban areas; and the generalized symmetric structure Tupu of hotspots is diverse. In addition, the results can reveal the hotspot formation mechanism and provide a reference for resource allocation.

**Keywords:** urban fringe area; Chengdu-Chongqing economic circle; spatially continuous wavelet transform; educational resources; density-field-based hotspot detector; Geo-information Tupu

## 1. Introduction

Urbanization is a double-edged sword. On the one hand, it can effectively promote the integrated development of urban and rural spaces. On the other hand, it has many impacts on the ecological environment, resource allocation, etc. [1–3]. Numerous studies have focused on urban and rural areas, from perspectives such as urban expansion [4,5], land cover change [6,7] and climate change [8,9]. However, these studies ignore the importance of urban fringe areas (UFAs). As the frontier of urbanization, a UFA has the characteristics of sensitivity, dynamics, and transition [10]. Currently, many factors restrict the healthy development of UFAs, such as land development and lack of management. In addition, UFAs can provide a new space for predicting the impact of urban form on the urban heat island effect [11] and optimizing the layout of urban infrastructure [12] in the future. Therefore, identifying UFAs has important practical implications for sustainable urban development.

In 1936, Herbert Louis proposed the concept of UFAs—the part of a rural area that is gradually occupied by urban built-up land [13]. Subsequently, due to the lack of data and the limited technology, scholars empirically defined UFAs as extending 10 to 50 km

from urban areas [14] or as a ring of 10 km width separating urban and rural areas [15]. Some scholars used statistical data to classify UFAs [16–18]. These methods were mainly limited by subjective factors and administrative boundaries, and their results often differed significantly from the true extent of UFAs. In recent years, the diversification of data sources and technology has promoted new methods for the identification of UFAs. These methods can be roughly summarized as the following steps.

First, select the characteristic indicators to identify UFAs. The characteristic indicators are mainly divided into the multi-indicator type and single-indicator type. The multi-indicator type involves selecting suitable factors from natural and socio-economic aspects to construct the indicator system [19–23], or applying deep learning to define the extent of UFAs [24]. The single-indicator includes land use information entropy [25–27], a comprehensive index of land use degree [28,29], land use dynamic degree [30], built-up land density [31], a light characteristics-combined value model [32–34], impervious surface index [35], density of point of interest [36], and population density [37]. Among them, each characteristic indicator has both strengths and weaknesses. Second, detect mutation points of the characteristic indicators. Common methods of detecting mutation points are the threshold method [38], Mann–Kendall method [39], wavelet transform method [40], and breakpoint method [41]. Notably, differences in the theoretical basis and applicability of mutation point detection methods can affect the results. Third, define the inner and outer boundaries of UFAs. Methods for determining the inner and outer boundaries mainly include the manual connection method [35], spatial superposition method [42], and Delaunay triangulation method [43]. In common, these defining processes do not avoid the influence of subjectivity. Finally, verify the identification results. At present, verifying the accuracy of the identification results is a challenge. Common methods of verification include remote sensing images, field surveys, landscape pattern index, night-time light (NTL), vegetation index, and population density [20,23,34]. Most of them are biased towards qualitative analysis.

UFAs are usually regarded as a key to healthy urban development. They face more complex problems in their development than urban and rural areas. Nevertheless, studies on the internal characteristics of UFAs and their differences between urban and rural areas are still relatively weak. Presently, some studies have focused on natural environmental issues in UFAs, such as land-use change, landscape patterns, ecological vulnerability, and ecological carrying capacity [44–48]. However, few studies have focused on the socio-economic issues in UFAs, especially those related to education. Some scholars indicated that there are many education problems in UFAs (e.g., compulsory education of migrant population's children cannot be guaranteed, the unbalanced spatial distribution of educational resources, and obvious disparities in educational outcomes) [49–51]. In these studies, the spatial extent of UFAs is based on administrative boundaries. However, as a product of urbanization, the spatial extent of UFAs should be in a state of dynamic change, rather than a traditional fixed administrative boundary. This demonstrates that taking the traditional administrative boundary as the true extent of UFAs to analyze the problem is not rigorous. Thus, it is necessary to study educational resources based on the identification of UFAs. Geo-information Tupu is the graphic thinking model, method, and tool for describing and understanding complex geo-phenomena and problems. It has a unique value in revealing the spatial pattern, evolution process, and element interaction mechanism of geoscience phenomena, and has the ability to reconstruct the geographical environment, evaluate the current situation, and predict the future [52]. Perhaps exploring the distribution of educational resources from a spatial perspective would be an effective way to achieve equity in education.

In summary, previous studies have many limitations. For characteristic indicators, the multi-indicator approach can comprehensively reveal the characteristics of UFAs, but the correlation between indicators will affect the accuracy of identification results; the single-indicator approach is not affected by correlation, but does not effectively reflect all the characteristics of UFAs. For mutation point detection, it is necessary to select the

optimal result through numerous comparative tests due to the theoretical differences in detection methods. For boundary definition, previous methods are subjective and have manual participation, which makes the results more uncertain. For results validation, using remote sensing images and field surveys are more reliable, but also more time-consuming; using NTL, vegetation index, etc., are hindered by some inaccuracies. To address these issues, we chose the Chengdu-Chongqing economic circle, the fourth-largest growth center of China, as the study area. We adopted the built-up land density, which closely reflects the characteristics of UFAs, as an indicator. Then, we used the gaus1-based spatially continuous wavelet transform to detect mutation points after comparison with other wavelet functions, and proposed a method that combines kernel density estimation, reclassification, and urban boundaries to define the inner and outer boundaries of UFAs. Finally, we employed two approaches (remote sensing images and NTL) to verify the reliability of the results. From an applied perspective, we selected the density-field-based hotspot detector and Geo-information Tupu to explore the spatial pattern of educational resources in UFAs. This study revealed the characteristics of urban development by exploring the spatial structure of UFAs and the spatio-temporal pattern of its resource elements.

## 2. Study Area and Data Sources

The Chengdu-Chongqing economic circle is located at the intersection of the Belt and Road and the Yangtze River Economic Belt, which has unique geographical advantages. As one of China's important strategic deployments, it is the region with the densest population, richest resources, fastest urbanization, and highest degree of openness in the west. In 2020, the urbanization rate of the Chengdu-Chongqing economic circle was only 55.74%, and it differed markedly from the other three major urban agglomerations in China. As early as 2016, the urbanization rates of the Pearl River Delta urban agglomeration, the Yangtze River Delta urban agglomeration, and the Beijing-Tianjin-Hebei urban agglomeration had already reached 84.49%, 61.44%, and 59.82% respectively. This suggests that a large proportion of UFAs created by urbanization in the future will be located in the study area. Hence, the Chengdu-Chongqing economic circle would be an ideal case study area for identifying UFAs. Figure 1 shows the scope of the study area. To ensure regional connectivity, Shizhu Tujia Autonomous County and Pengshui Miao Tujia Autonomous County were included in the study region.

Data selected for the study included administrative divisions, land use, points of interest (POI), urban boundaries, and NPP-VIIRS-like NTL (National Polar-Orbiting Partnership Visible Infrared Imaging Radiometer Suite like night-time light). In 2020, we obtained the POI data from Amap (Amap is a leading provider of digital map content, navigation, and location services). After repeated data cleaning and coordinate correction, we collected 2007 valid data points (which included both basic education resources and higher education resources). Of these, Chengdu contained 1145 data points, and Chongqing contained 862 data points. The coordinate reference system of all data is the WGS-84 coordinate system. Table 1 shows further information about the data.

**Table 1.** Datasets used in the study.

| Dataset | Time | Data Type | Spatial Resolution | Data Source |
|---|---|---|---|---|
| Administrative divisions | 2020 | Vector | None | https://www.webmap.cn, accessed on 10 February 2021 |
| Land use | 2020 | Raster | 30 m | http://www.globallandcover.com, accessed on 11 May 2021 |
| POI | 2020 | Vector | None | https://lbs.amap.com, accessed on 26 December 2020 |
| Urban boundaries | 2018 | Vector | None | http://data.ess.tsinghua.edu.cn, accessed on 10 September 2021 |
| NPP-VIIRS-like NTL | 2020 | Raster | 500 m | https://doi.org/10.7910/DVN/YGIVCD, accessed on 18 June 2021 |

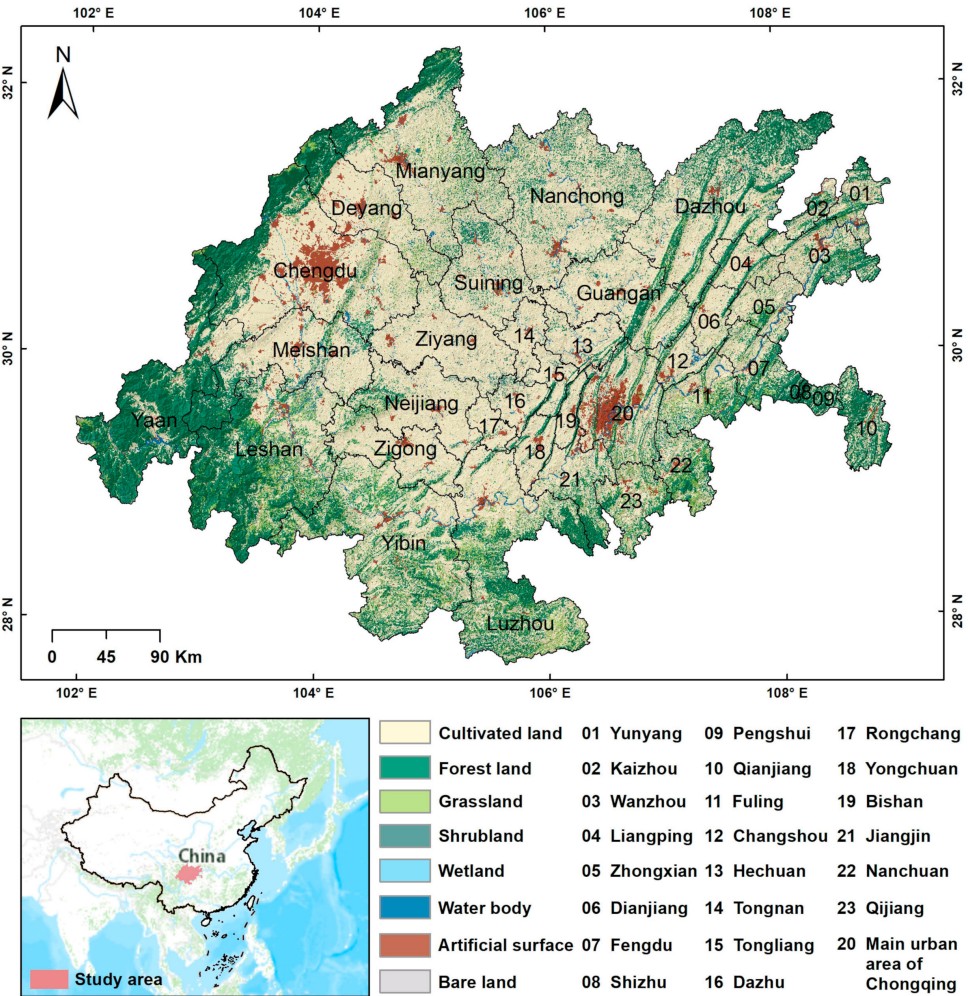

**Figure 1.** Location of the study area and spatial distribution of land use types.

## 3. Methods

In this study, we identified UFAs by selecting indicators and grid scales, detecting mutation points, and defining the extent of UFAs (Sections 3.1–3.3). Section 5.1 is the verification and evaluation of UFA identification results. Then, we explored the spatial pattern of educational resources by the density-field-based hotspot detector and the generalized symmetric structure Tupu (Section 3.4). Figure 2 shows the whole workflow.

### 3.1. Selection of Indicators and Grid Scales

Combined with previous studies on UFA identification and modern urban spatial structure theory, it was found that land use and socio-economic characteristics can effectively divide the spatial extent of UFAs [53,54]. Under the dual influences of urban and rural areas, the land use in UFAs appears disorderly. Urban areas maintain higher values of built-up land density; the value of built-up land density in UFAs fluctuates; and rural areas maintain lower values of built-up land density. Therefore, this study selected built-up land density as an indicator to identify UFAs. First, we reclassified the land use data at 30 m spatial resolution into built-up land and non-built-up land. Then, we extracted the built-up land and resampled it into four different spatial resolutions (250 m, 500 m, 750 m, and 1000 m). Finally, we used the fishnet tool in ArcGIS to quantify the built-up land density (the proportion of built-up land area in the grid to the total area of the grid) at 250 m, 500 m, 750 m, and 1000 m spatial resolutions.

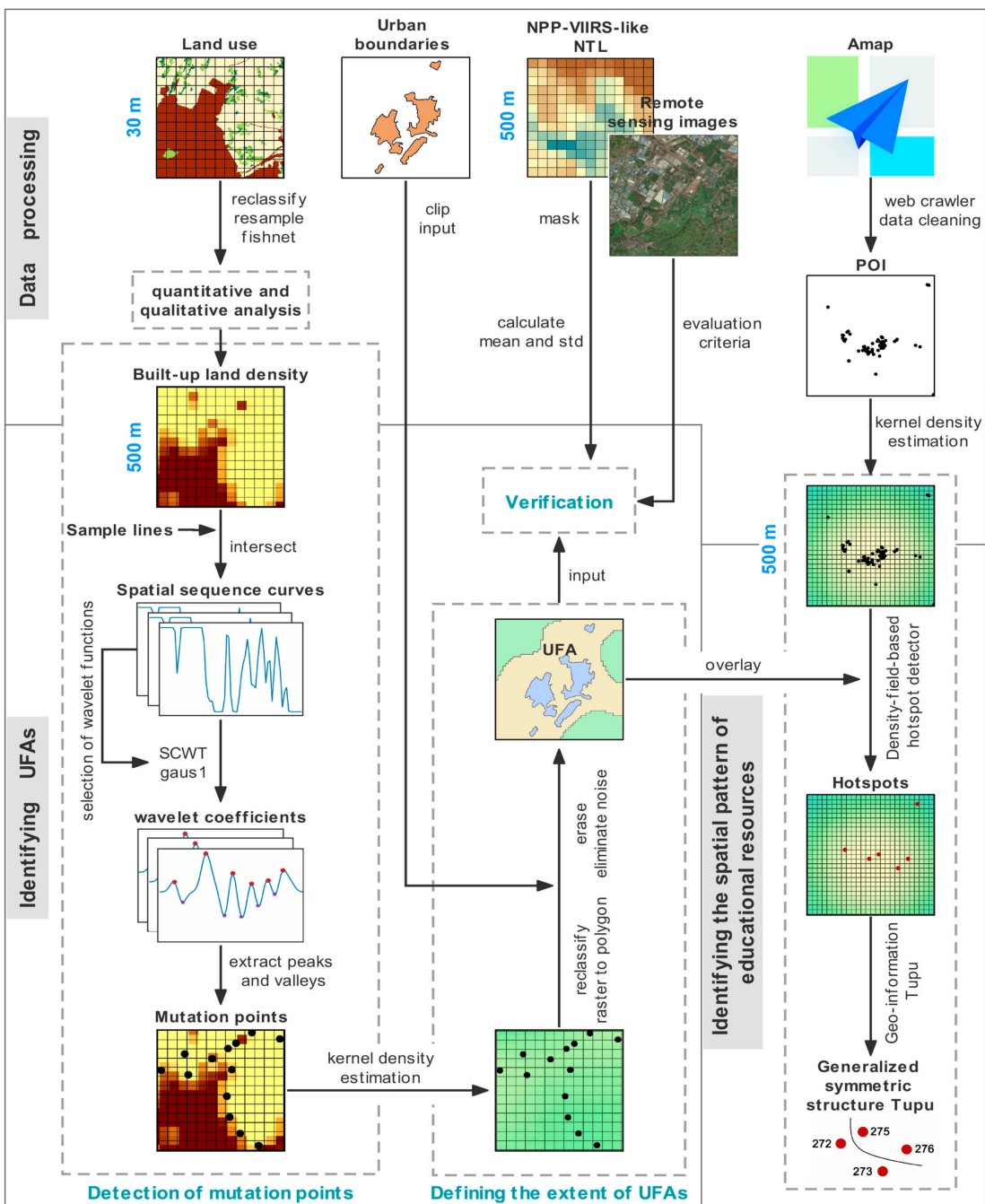

**Figure 2.** Workflow of this study.

How to determine the appropriate grid-scale has been an extremely important part of geographic research. Spatial heterogeneity occurs when variables are unevenly distributed in space. Moreover, it has different manifestations at different grid scales. As grid-scale changes, the impact of spatial heterogeneity on the research results also changes [55]. Thus, it is necessary to select the best grid-scale prior to carrying out the study. Previous studies have mainly used a single approach to determine the grid-scale [27,43], but with results that are not convincing. We used a combination of quantitative and qualitative methods to determine the grid-scale, which we consider to be more reasonable. First, we chose the kriging interpolation model of the semi-variogram to quantitatively describe the structure and randomness of the regional variable (built-up land density), and utilized the band collection statistics tool in ArcGIS to determine the correlation coefficient between the

predicted result and the original data. Equation (1) is the mathematical expression of the semi-variogram. Table 2 shows the computed results of parameters at different scales.

$$\omega(d) = \frac{1}{2N(d)} \sum_{i=1}^{N(d)} [Z(x_i) - Z(x_i + d)]^2,$$ (1)

where $\omega(d)$ represents the semi-variogram; $d$ is the interval distance between sample points; $N(d)$ is the logarithm of all observation points at an interval of $d$; and $Z(x_i)$ and $Z(x_i + d)$ are the built-up land density at $x_i$ and $x_i + d$ respectively.

**Table 2.** Parameters of the semi-variogram at different grid scales.

| Scale/m × m | $C_0$ | $C + C_0$ | $C_0/(C + C_0)$ | A | r |
|---|---|---|---|---|---|
| 250 | 0.0026 | 0.0189 | 0.14 | 825 | 0.9103 |
| 500 | 0.0042 | 0.0119 | 0.26 | 1542 | 0.8539 |
| 750 | 0.0019 | 0.0113 | 0.14 | 1785 | 0.8250 |
| 1000 | 0.0021 | 0.0140 | 0.13 | 2245 | 0.7989 |

Note: $C_0$ is the nugget variance, which means random error; $C + C_0$ is the sill, which means the degree of spatial heterogeneity; the ratio $C_0/(C + C_0)$ quantifies the nugget effect, which means spatial correlation, and a larger $C_0/(C + C_0)$ represents weaker spatial correlation; A is the range, which means the maximum distance of spatial autocorrelation; and r is the correlation coefficient, which quantifies the consistency between the predicted result and the original data.

As shown in Table 2, the differences in $C_0$ and $C + C_0$ at different grid scales are small, so this study mainly focuses on the correlation coefficient and the nugget effect. Larger correlation coefficients and nugget effects indicate that the grid-scale is more applicable. Evidently, the grid scales of 250 m and 500 m are more suitable. Subsequently, we qualitatively examined the differences between different grid scales by extracting the built-up land density along the same transect. At 250 m grid-scale, the loss of original information is small, and the data redundancy is large (Figure 3); at 1000 m grid-scale, the data redundancy is small, and the original information loss is large; the 500 m and 750 m grid scales reduce data redundancy but also preserve the characteristics of the original information. Combining the results of quantitative and qualitative analysis, we finally decided to use 500 m as the grid-scale.

### 3.2. Detection of Mutation Points
### 3.2.1. Spatially Continuous Wavelet Transform

The continuous wavelet transform is widely used in signal analysis, medical imaging, and image processing [56,57]. Its principle is to decompose the original function by the inner product of the wavelet function and signal. The continuous wavelet transform is also a common method for detecting mutation points. In the detection, the first derivative of a smooth function is generally taken as a wavelet function for the wavelet transform. After the wavelet transform, the mutation points will correspond to the modulus maximum points of the wavelet coefficients [40]. The spatially continuous wavelet transform regards the signal as a data sequence sampled from geographic spatial elements. The calculation is shown in Equation (2):

$$SCWT = s(x)\varphi(x) = \frac{1}{\sqrt{a}} \int_{-\infty}^{+\infty} s(x)\varphi\left(\frac{x - \tau}{a}\right) dx,$$ (2)

where $SCWT$ represents the wavelet coefficients; $a$ is the scale factor; $s(x)$ is the spatial distribution of built-up land density; $\varphi(x)$ is the wavelet function; and $\tau$ is the shift factor.

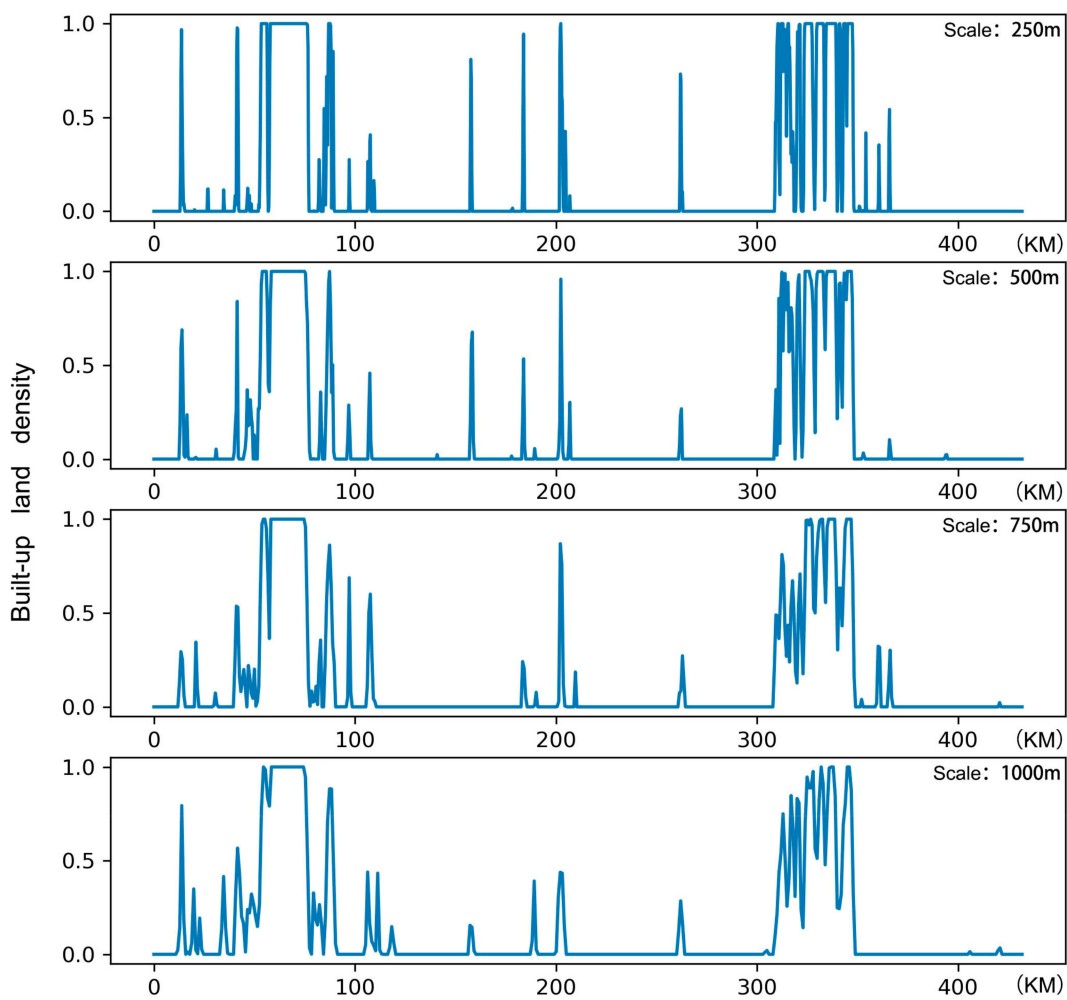

**Figure 3.** Changes of built-up land density at different grid scales (taking the built-up land density of the 216° transect as an example).

Regarding the selection of wavelet functions, we compared the mutation points extracted by seven wavelet functions on the same transect. Specifically, these were the Gaussian wavelet (gaus1), Mexican Hat wavelet (mexh), Morlet wavelet (morl), Complex Gaussian wavelet (cgau1), Shannon wavelet (shan), Frequency B-spline wavelet (fbsp), and Complex Morlet wavelet (cmor). Figure 4 reveals obvious differences in the results of mutation points extracted by different wavelet functions. Among them, the mutation point extracted by the gaus1-based spatially continuous wavelet transform is most consistent with the position where the density of the built-up land changes. Consequently, we selected gaus1 to detect mutation points.

### 3.2.2. Detection Process

Taking the center of Chengdu as the origin, we constructed a circular region as an example to detect mutation points. We determined the center point by the centroid of the largest patch in the urban boundaries. The detection process comprised the following steps. (1) Extraction of spatial sequence curves of characteristic indicators. As shown in Figure 5, we intersected the sampling line layer with the built-up land density layer. The sample line layer contained 360 sample lines (the due north direction is the 0° transect, and the degrees increased clockwise). (2) Spatially continuous wavelet transform. As shown in Figure 6, we performed the spatially continuous wavelet transform on the original signal. The position of the mutation points (red solid circles and purple pentagrams) is consistent with the original signal. (3) Spatial visualization of mutation points. We mapped the mutation

points into space by latitude and longitude. As shown in Figure 7, the mutation points extracted by spatially continuous wavelet transform based on gaus1 are accurate (locations with large variations in build-up land density).

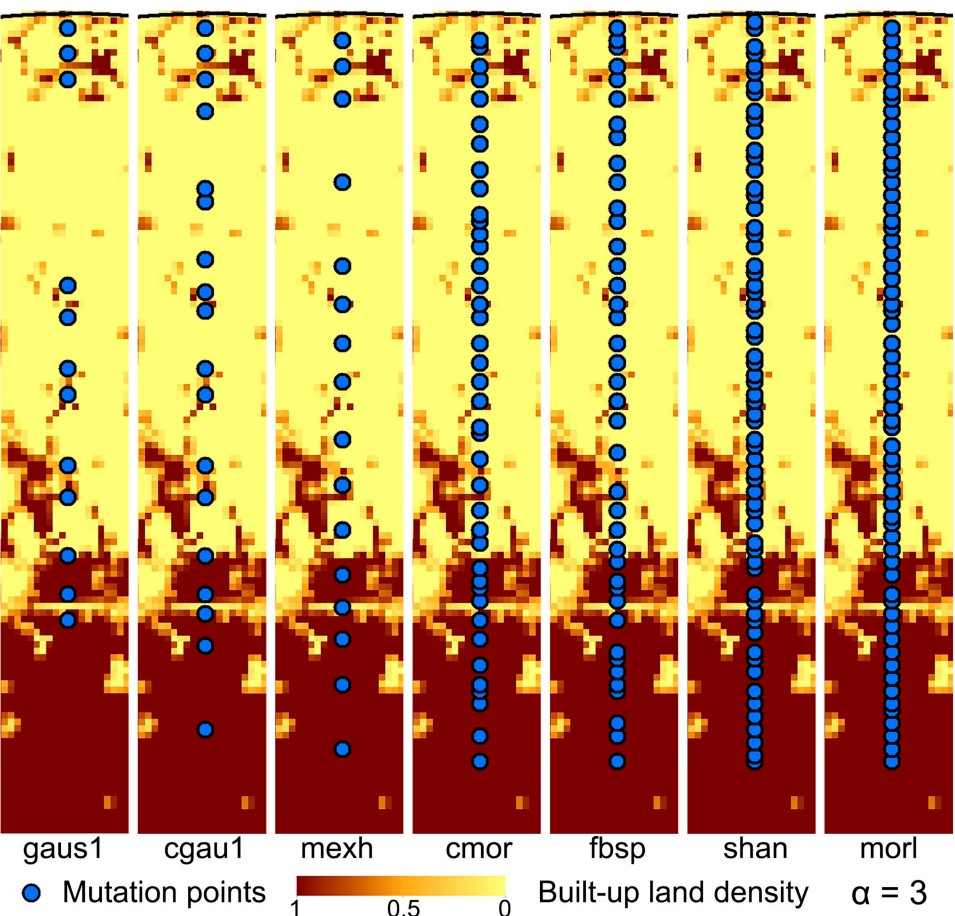

**Figure 4.** Extraction results with different wavelet functions (taking the built-up land density of the 0° transect as an example).

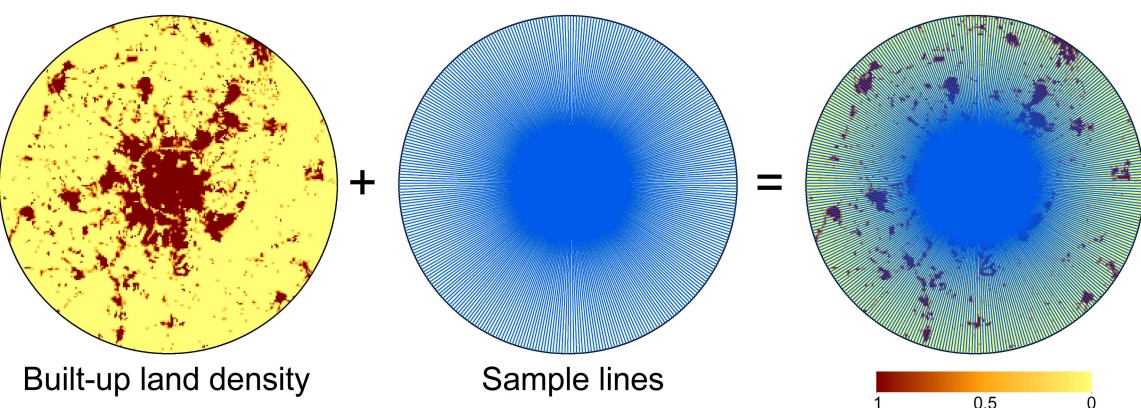

**Figure 5.** Extraction process of built-up land density.

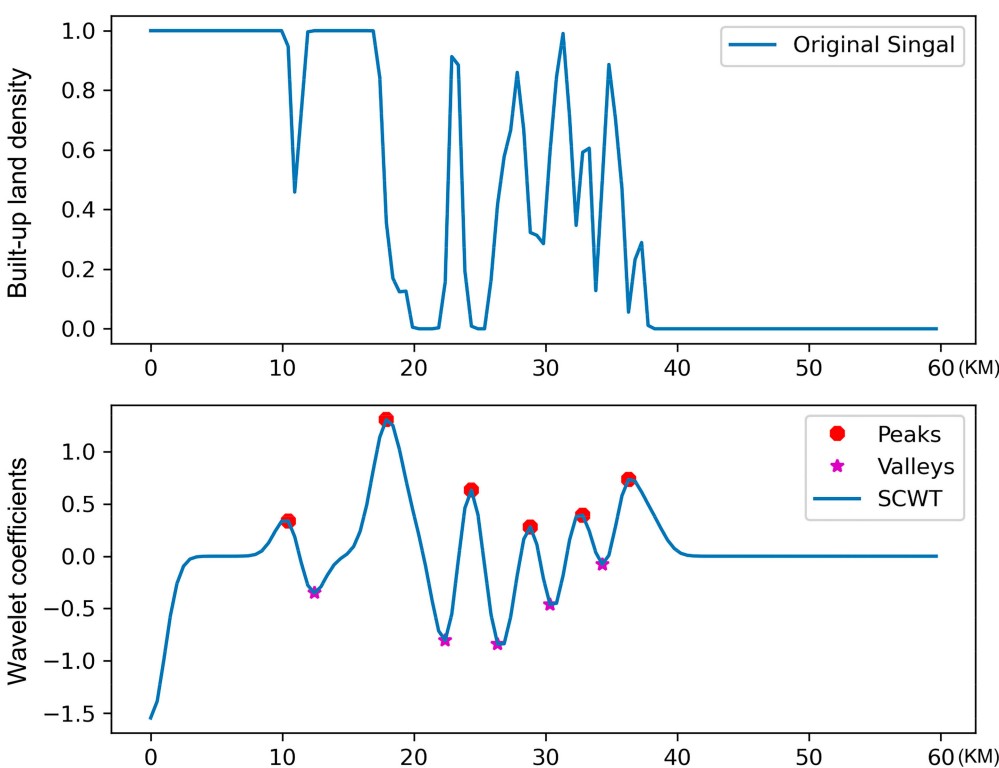

**Figure 6.** Process of spatially continuous wavelet transform (taking the built-up land density of the 216° transect as an example).

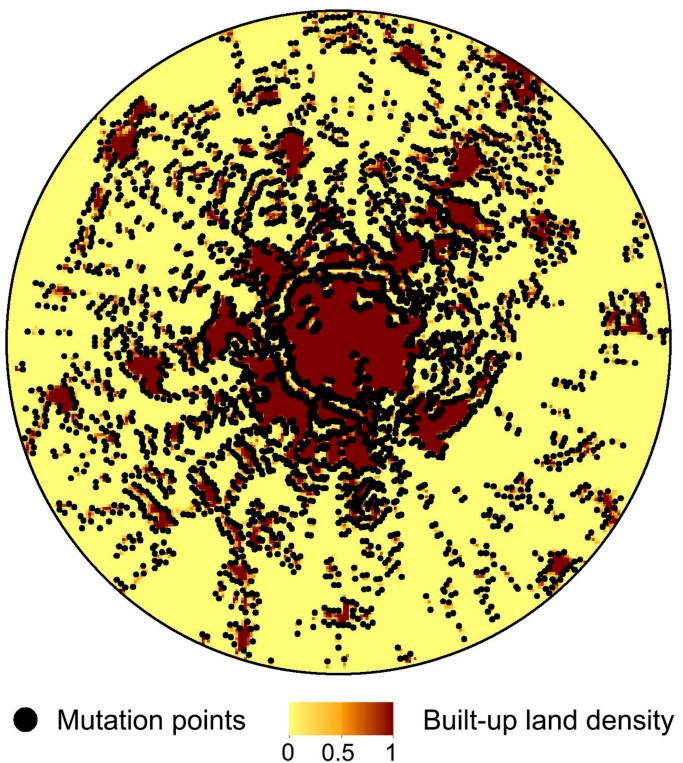

**Figure 7.** Extraction results for the sample. The sample area is located between longitudes 103°15′ E to 104°45′ E and latitudes 30°00′ N to 31°15′ N. It is the most developed area of Chengdu and its urbanization is evident. Furthermore, the radius of the sample area is approximately 60 km.

### 3.3. Defining the Extent of UFAs

In Figure 7, urban and rural areas mostly do not contain mutation points. In this case, kernel density estimation can generate a density surface with boundary effects [58]. Therefore, we divided the inner and outer boundaries of UFAs by the kernel density estimation and carried out a reclassification. The calculation is shown in Equation (3):

$$\hat{f}(x,y) = \frac{3}{nh^2\pi} \sum_{i=1}^{n} \left[ 1 - \frac{(x-x_i)^2 + (y-y_i)^2}{h^2} \right]^2,$$

(3)

where $\hat{f}(x,y)$ is the value of the kernel density at the spatial position $(x,y)$; $h$ is the search radius as determined by the bandwidth estimation formula in Silverman's empirical rule [59], which can effectively prevent spatial outliers; $(x_i, y_i)$ are the coordinates of the mutation point $i$; $n$ is the number of mutation points whose distance from position $(x_i, y_i)$ is less than or equal to $h$; $x$ and $y$ represent the coordinates of the center point of the grid to be estimated within the search radius; $(x-x_i)^2 + (y-y_i)^2$ is the square of Euclidean distance between the center point of the grid to be estimated and the mutation point $i$ within the search radius.

Figure 8 summarizes the process of defining the extent of the UFA. Firstly, we estimated the kernel density of the mutation points (Step 1 in Figure 8). Secondly, we classified the kernel density values into two categories, by reclassification (Step 2 in Figure 8). The results of Step 2 reflected that the kernel density estimation and reclassification cause inaccuracies in the inner boundaries, which arise due to the strong influence of surrounding points on the interior during kernel density estimation. Thirdly, we selected the urban boundary data instead of the inner boundary obtained in Step 2 (Step 3 in Figure 8). Finally, we obtained the spatial extent of the UFA by using the erase tool (Step 4 in Figure 8). When defining the inner and outer boundaries, the reclassification method can identify noise points. Thus, we manually deleted some noise points, as appropriate.

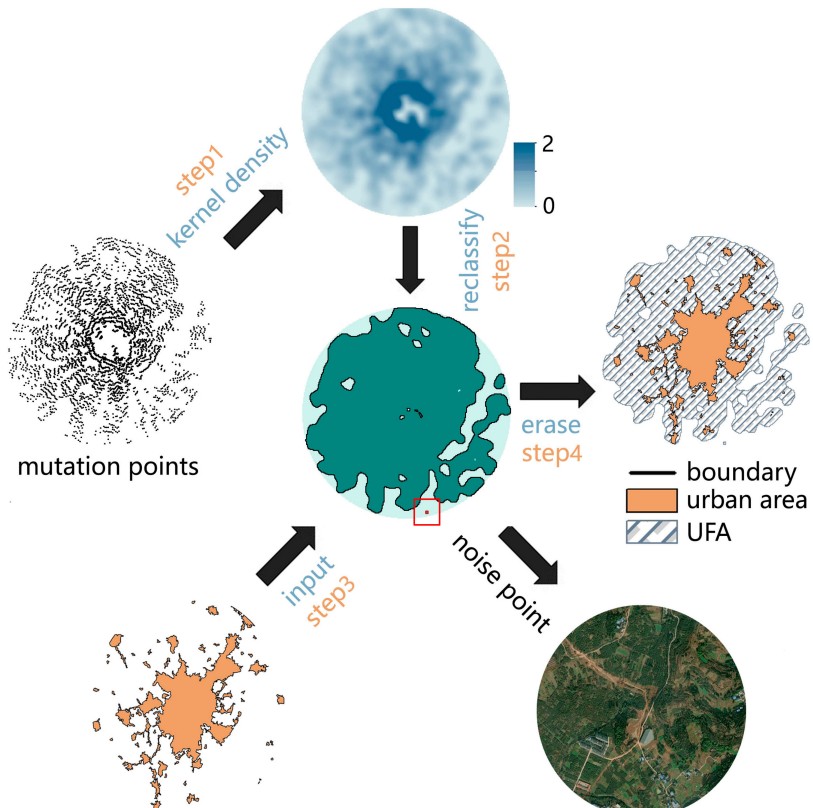

**Figure 8.** Process of defining the extent of UFAs.

### 3.4. Analysis Method of Spatial Patterns

3.4.1. Density-Field-Based Hotspot Detector

Typically, the density surface can express the density distribution characteristics of elements in space, but it is difficult to identify hotspots. This study chose the density-field-based hotspot detector [60] to extract the hotspots of educational resources in UFAs. To maintain a consistent grid-scale, we generated the density surface of educational resources at the 500 m grid-scale based on POI data and kernel density estimation, and calculated its extremum of neighborhood pixels through neighborhood statistics. Notably, we optimized the density-field-based hotspot detector (by eliminating zero values) when obtaining the extremum of neighborhood pixels. Eliminating zero values can effectively avoid the appearance of abnormal hotspots when the density values of multiple neighboring pixels are zero. Then, we obtained a non-negative surface by subtracting the density surface of educational resources from the surface of extreme values. In the non-negative surface, we used reclassification to extract zero values (zero values represent hotspots). Finally, we extracted the original kernel density of educational resources by using the hotspots, and classified the density according to the Jenks natural breaks classification method [61]. Figure 9 shows the process of detecting hotspots.

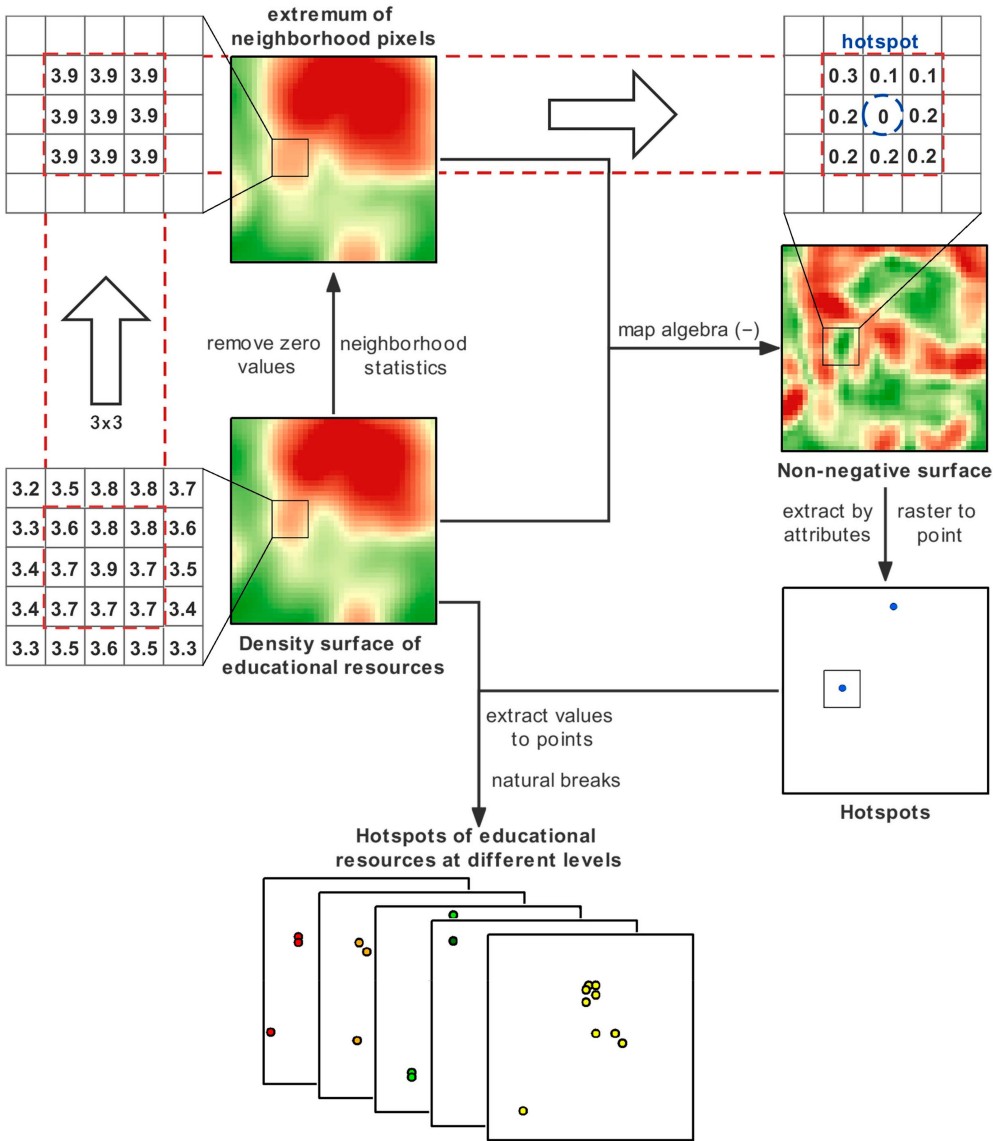

**Figure 9.** Process of detecting hotspots.

3.4.2. Generalized Symmetric Structure Tupu

A Geo-information Tupu is often used to describe the characteristics of socio-economic and natural processes in time and space. Not only is a Tupu well suited to image expression, but it can also simplify complex phenomena [62]. Currently, scholars use a Geo-information Tupu to analyze the temporal and spatial changes of land use [63,64], the dynamic changes of landscape [65], and the spatial structure of catering services [66]. A generalized symmetric structure Tupu is a kind of Geo-information Tupu, which mainly emphasizes the symmetry relationships of objects in space. In this study, we selected a variety of symmetrical structures (Figure 10) to reveal the spatial pattern of educational resources in UFAs.

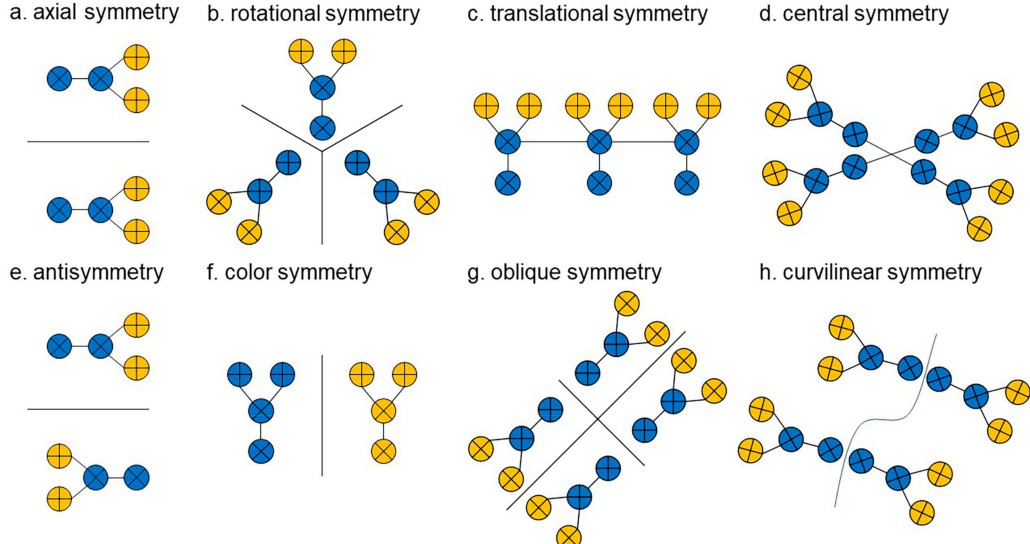

**Figure 10.** Generalized symmetric structure Tupu. Parts (**a**–**h**) represent 8 kinds of symmetrical structures.

## 4. Results

### 4.1. Identification Results of UFAs

Based on the method proposed in this study, we identified the UFAs of each city in the Chengdu-Chongqing economic circle in 2020 and divided the city into three parts (referring to the urban area, UFA, and rural area in Figure 11) according to their inner and outer boundaries. Chongqing is a municipality directly under the central government. Furthermore, it belongs to the provincial administrative unit. Consequently, the districts and counties under the jurisdiction of Chongqing should belong to municipal administrative units. However, it is not reasonable to analyze the UFA according to the municipal administrative units, because the area of each district and county in Chongqing is not equal to that of other cities in the Chengdu-Chongqing economic circle. By considering the regional area and the development plan of Chongqing, we divided Chongqing into three parts: the main urban area (A1), the main urban new area (A2), and other areas (A3).

As shown in Figure 11, UFAs contain the following characteristics. (1) Spatial morphological characteristics. UFAs are generally distributed at the periphery of urban areas (in the shape of a ring) and have the distinctive features of varying widths and a crisscrossing relationship with rural areas. (2) Spatial distribution characteristics. UFAs of different cities have different relative locations (e.g., UFAs of Neijiang and Zigong are located in the east; UFAs of Ziyang and Mianyang are located in the west). In the Chengdu-Chongqing economic circle, the UFAs of each city are mainly distributed in the central and northern parts of the cities. (3) Spatial structure characteristics. Chengdu and A1 have a single-core structure, and the others have a multi-core structure.

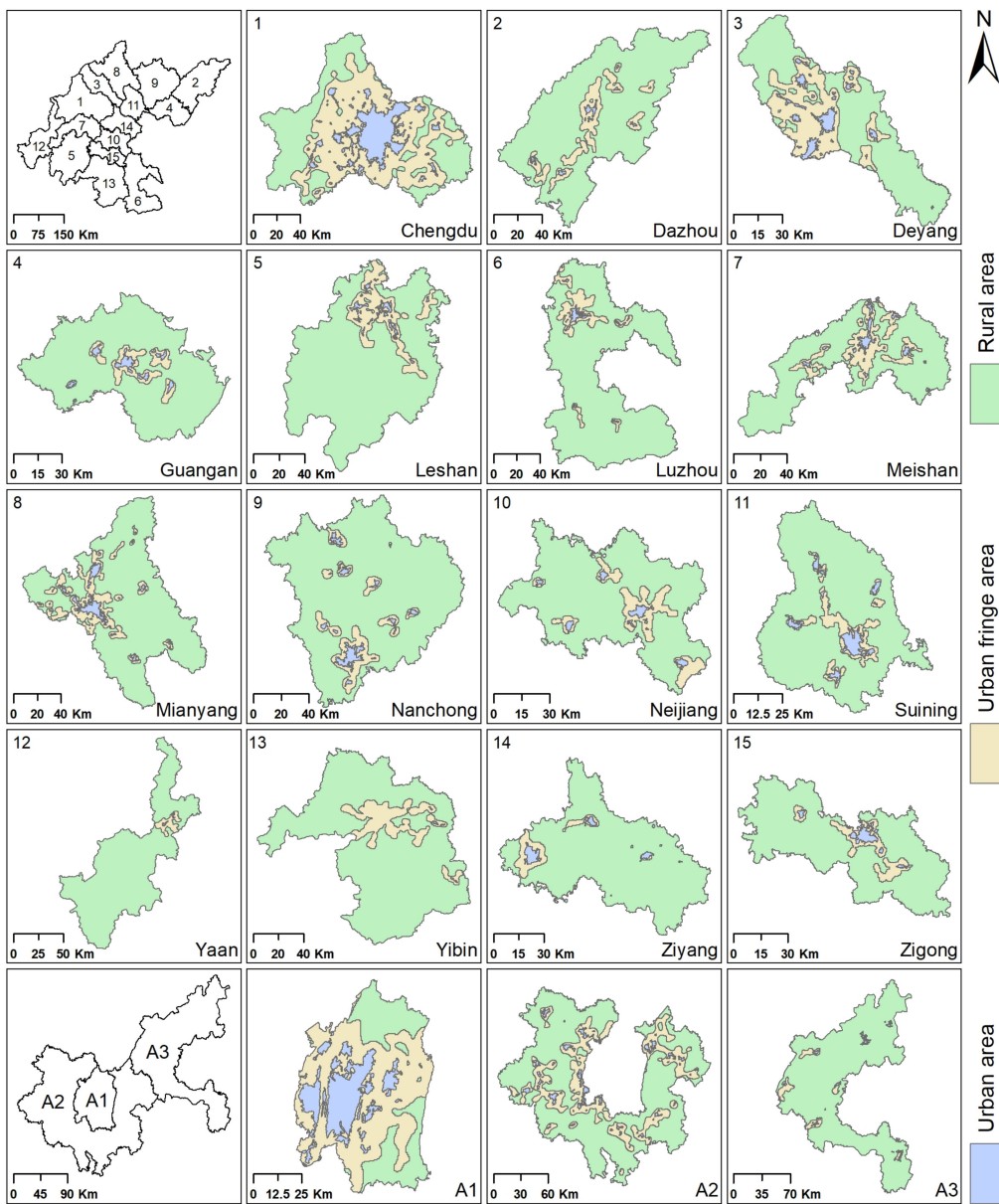

**Figure 11.** UFA identification results.

We calculated the areal proportions of urban areas, UFAs, and rural areas, to facilitate analysis of the development level of each city. The proportion of UFAs to total city areas can reflect the rate of urbanization. Hence, we divided the urbanization characteristics into three types, according to the respective proportions. Those below 10% were classified as the slow development type; proportions between 10% and 30% were classified as the stable development type; and those above 30% were classified as the fast development type. As shown in Figure 12, Chengdu and A1 are fast development types, with UFA proportions of 47.98% and 53.79%, respectively. The UFA formed by fast-developing regions is relatively continuous. As the two cores of the Chengdu-Chongqing economic circle, Chengdu and A1 help drive the development of surrounding areas. A2, Deyang, and Meishan are stable development types, with the UFA proportions 23.56%, 29.58%, and 17.55% respectively. The UFA formed by stable-developing regions is relatively scattered, and contains many urban sub-centers. Ya'an, Ziyang, and A3 are slow development types, with UFA proportions 2.96%, 5.14%, and 3.49% respectively. Their less desirable geographical locations may be the main reason for their slow development. Overall, the Chengdu-Chongqing economic circle contains 2 fast-developing regions, 8 stable-development regions, and 8 slow-developing

regions. This indicates a good development level of the Chengdu-Chongqing economic circle, but guidance from the two cores is still needed to drive the synergistic development between regions.

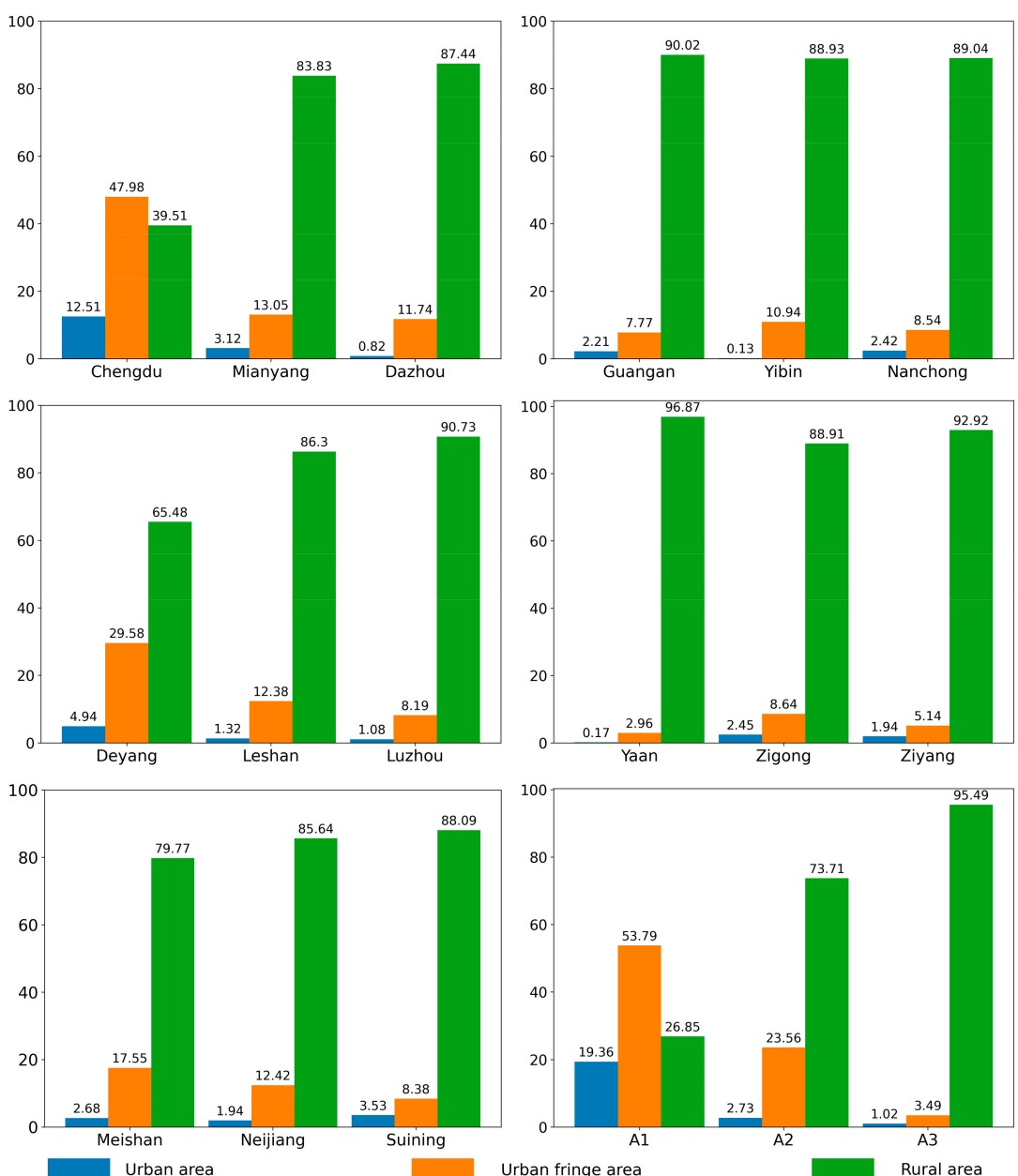

**Figure 12.** Areal proportions of urban areas, UFAs, and rural areas. Area of urban areas, UFAs and rural areas are shown in Table 3.

**Table 3.** Area of urban areas, UFAs and rural areas. Areas are retained to two decimal places.

| City | Urban Area (Km$^2$) | UFA (Km$^2$) | Rural Area (Km$^2$) | Total Area (Km$^2$) |
|---|---|---|---|---|
| Chengdu | 1799.33 | 6878.71 | 5663.34 | 14,335.38 |
| Mianyang | 356.21 | 1490.25 | 9573.38 | 11,419.84 |
| Dazhou | 102.26 | 1471.24 | 10,958.59 | 12,532.09 |
| Deyang | 293.79 | 1758.69 | 3893.26 | 5945.74 |
| Guang'an | 140.54 | 493.73 | 5723.20 | 6357.47 |

**Table 3.** *Cont.*

| City | Urban Area (Km²) | UFA (Km²) | Rural Area (Km²) | Total Area (Km²) |
|---|---|---|---|---|
| Leshan | 167.64 | 1576.59 | 10,990.96 | 12,735.19 |
| Luzhou | 132.51 | 1002.55 | 11,104.13 | 12,239.19 |
| Meishan | 191.42 | 1250.97 | 5686.77 | 7129.16 |
| Nanchong | 301.27 | 1064.58 | 11,101.78 | 12,467.63 |
| Neijiang | 103.75 | 666.07 | 4591.95 | 5361.77 |
| Suining | 188.06 | 446.02 | 4687.61 | 5321.69 |
| Ya'an | 16.08 | 282.17 | 9241.35 | 9539.60 |
| Yibin | 17.12 | 1449.95 | 11,791.93 | 13,259.00 |
| Ziyang | 111.76 | 295.59 | 5339.00 | 5746.35 |
| Zigong | 107.05 | 377.99 | 3887.75 | 4372.79 |
| A1 | 1058.65 | 2940.66 | 1467.66 | 5466.97 |
| A2 | 632.75 | 5466.12 | 17,096.49 | 23,195.36 |
| A3 | 181.17 | 617.66 | 16,902.06 | 17,700.89 |

*4.2. Spatial Pattern of Educational Resources*

4.2.1. Spatial Distribution of Hotspots

This study explored the spatial pattern of educational resources in two representative UFAs (UFAs of Chengdu and Chongqing). By using the density-field-based hotspot detector, we identified 117 hotspots in Chengdu and 224 hotspots in Chongqing. Then, we used the Jenks natural breaks classification method to divide hotspots into 5 levels (labelled I-V, representing small, medium-small, medium, medium-large and large educational resources hotspots, respectively) [60,66]. The specific information and spatial distribution are shown in Table 4, Figures 13 and 14.

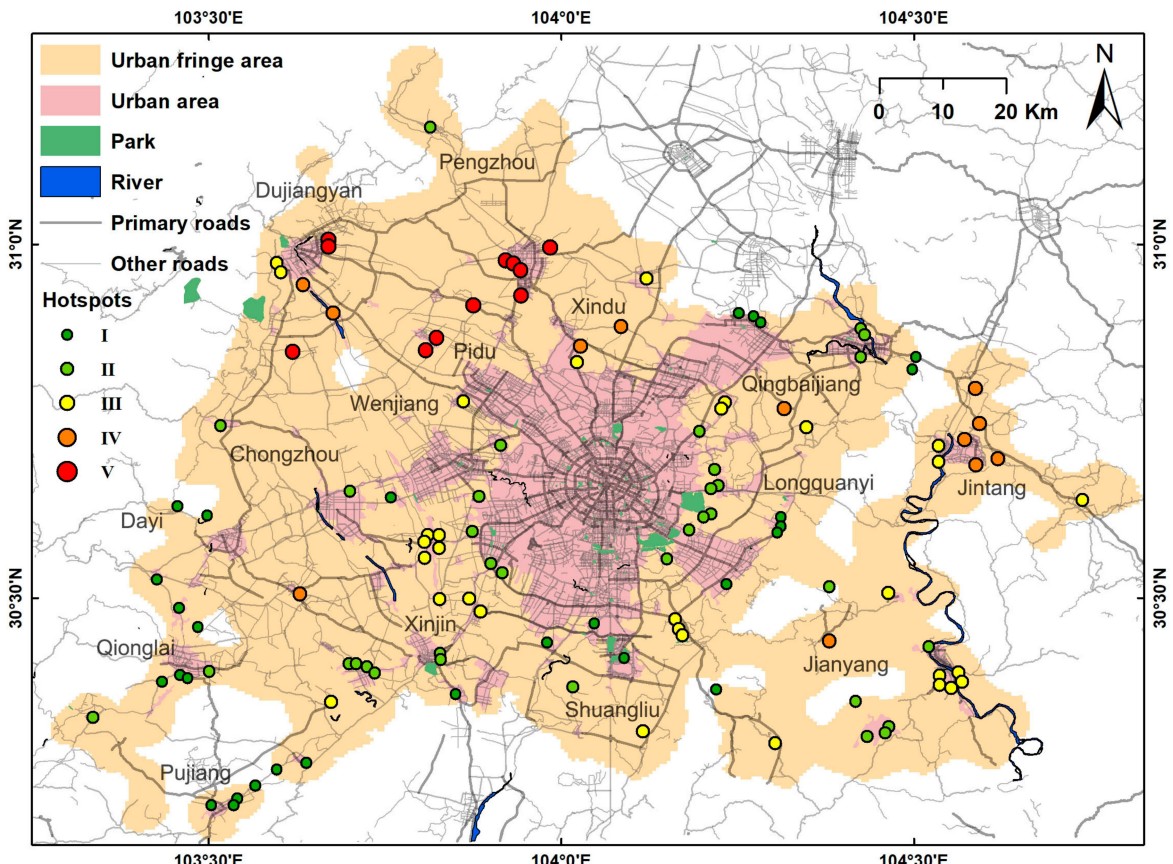

**Figure 13.** Results of hotspot detection (UFAs of Chengdu).

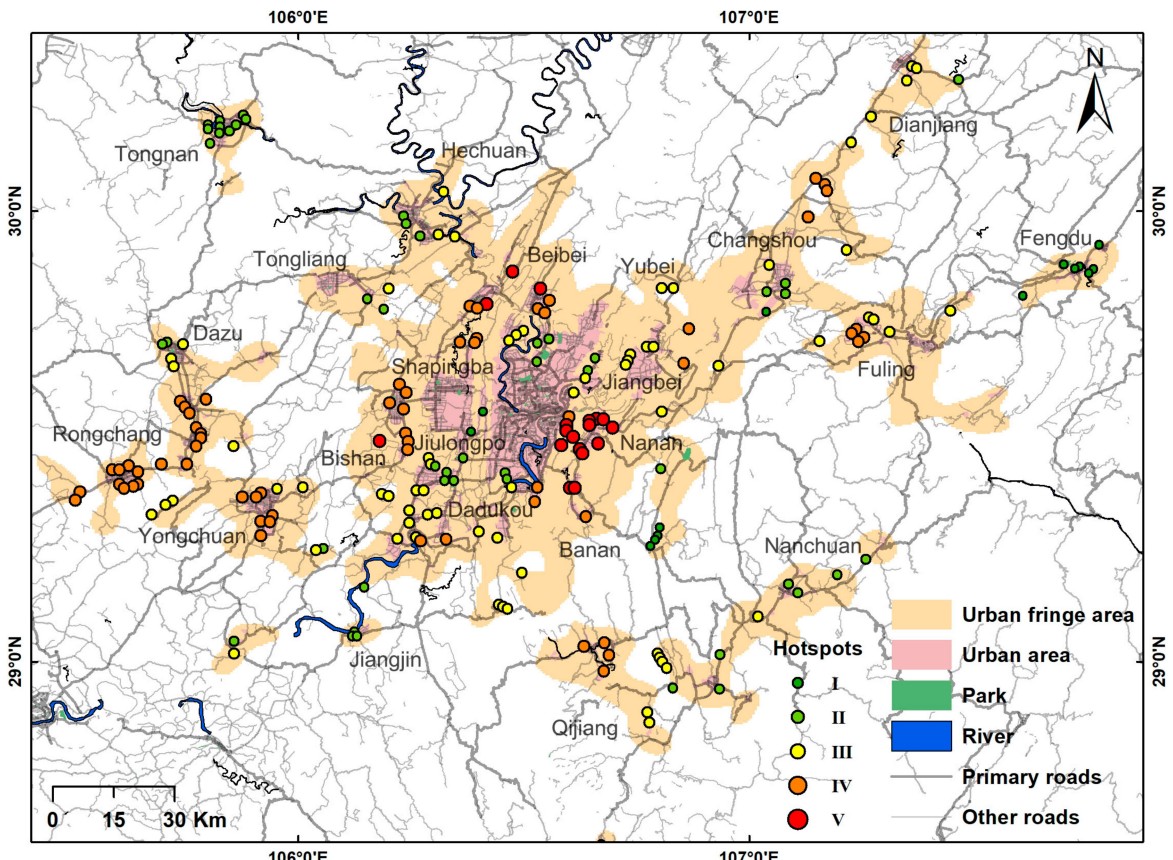

**Figure 14.** Results of hotspot detection (UFAs of Chongqing).

**Table 4.** Basic information on hotspots of educational resources at different grades.

| Region | Grade | Number of Hotspots | Proportion | Mean Density |
|---|---|---|---|---|
| UFAs of Chengdu | I | 29 | 24.78% | 0.048 |
| | II | 34 | 29.06% | 0.096 |
| | III | 31 | 26.50% | 0.14 |
| | IV | 12 | 10.26% | 0.21 |
| | V | 11 | 9.40% | 0.31 |
| UFAs of Chongqing | I | 23 | 10.26% | 0.009 |
| | II | 54 | 24.12% | 0.029 |
| | III | 67 | 29.91% | 0.05 |
| | IV | 62 | 27.68% | 0.075 |
| | V | 18 | 8.03% | 0.106 |

Table 4 shows the basic information on hotspots of educational resources at different grades. In the UFAs of Chengdu, hotspots with grades I to III account for a large proportion, and hotspots with grades IV to V account for a small proportion. This indicates that the hotspots of educational resources in the region are mainly small to medium. In UFAs of Chongqing, hotspots with grades II to IV account for a large proportion, and hotspots with grades I and V account for a small proportion. This demonstrates that the hotspots of educational resources in that region are dominated by medium-small to medium-large. In general, the number of hotspots in the UFAs of Chengdu is low, but their mean density is high, and the number of hotspots in the UFAs of Chongqing is large, but their mean density is low. This contrast may arise because Chongqing is located in a mountainous area, whereas Chengdu is on the plains. In mountainous areas, educational resources are scattered and difficult to gather into large hotspots. In the plains, the pattern is the opposite.

As shown in Figure 13, the hotspots (UFAs of Chengdu) of grade IV to V are mainly distributed in the north, and the hotspots of grade I to III are mainly distributed in the center and south. The distribution of hotspots follows the traffic-intensive areas or proximity to urban areas. Hotspots with a higher grade tend to be distributed in better-developed areas, which further suggests that the economy can boost the development of the education sector. As shown in Figure 14, the hotspots (UFAs of Chongqing) of grade IV to V are mainly distributed around the main urban area, and the hotspots of grade I to III are scattered in disparate regions. Other distribution patterns of hotspots are similar to the UFAs of Chengdu.

### 4.2.2. Generalized Symmetric Structure Tupu of Hotspots

To facilitate analysis, we numbered the hotspots in UFAs of Chengdu from 1 to 117, and hotspots in UFAs of Chongqing from 118 to 341. Since there were many symmetrical structures, it was difficult to discuss all the structures. Hence, we selected some representative structures for analysis. Figures 15 and 16 show the extraction results of the generalized symmetric structure Tupu for hotspots. The formation of the curvilinear symmetry structure (Figures 15a and 16a) may be due to spatial heterogeneity; there are different natural conditions on both sides of the symmetry axis, such as mountains and rivers, which limit the balanced development of educational resources. The formation of antisymmetry structures (Figures 15b and 16b) may be greatly affected by policy; this structure corresponds to the shape and trend of urban development. The formation of the color symmetry structure (Figures 15d and 16c) is mainly controlled by the level of economic development, since the unbalanced development on both sides of the symmetry axis caused differences in the grades of hotspots. The formation of the axial symmetry structure (Figure 15c) and central symmetry structure (Figure 16d) shows that the traffic, terrain, and other conditions around the axis are similar. The rotational symmetry structure (Figures 15e and 16e) reveals that the hotspots of the corresponding levels in the two regions are developing in three directions: north, southwest, and southeast. Combined with the distribution of other hotspots, it can be inferred that the driving force for the development of educational resources in a southwards direction is weak.

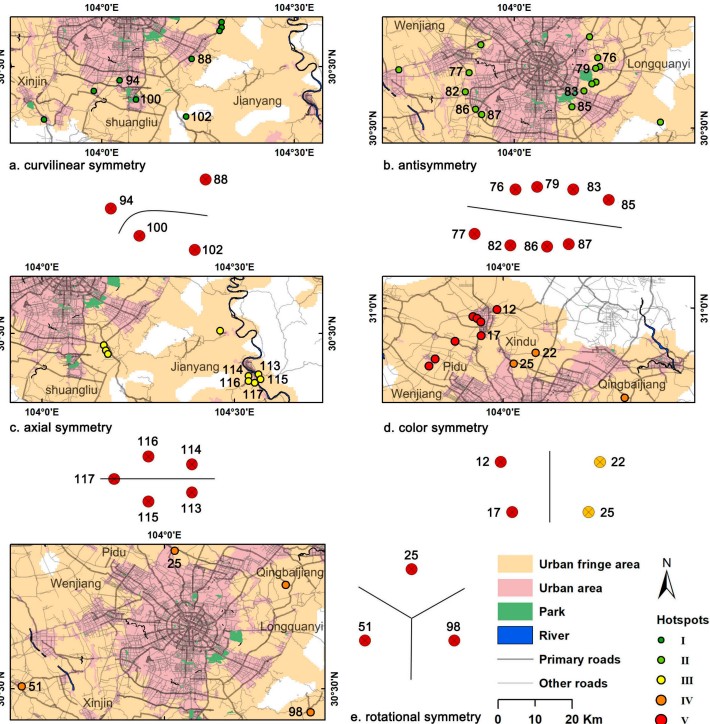

**Figure 15.** Extraction results of the generalized symmetric structure Tupu for hotspots (UFAs of Chengdu).

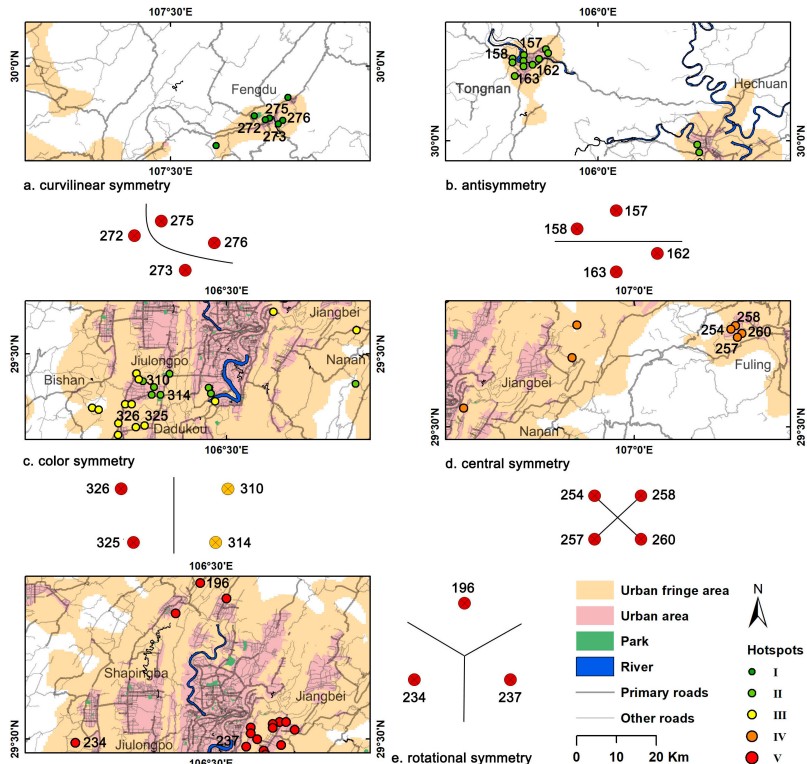

**Figure 16.** Extraction results of the generalized symmetric structure Tupu for hotspots (UFAs of Chongqing).

## 5. Discussion

### 5.1. Verification and Evaluation of UFA Identification Results

Recent studies have mostly used NTL, vegetation index, landscape pattern index, etc., to verify their results [20,23]. However, these methods are biased towards qualitative analysis and cannot accurately measure the reliability of the identification. Thus, we chose two methods (NPP-VIIRS-like NTL and remote sensing images) to verify the accuracy of the identification results.

NTL can effectively reflect the spatial distribution characteristics of the population and economy [67,68]. There are significant demographic and economic differences between urban areas, UFAs, and rural areas. The accuracy of identification results can be verified from the macro-scale through the NTL. Due to the short available time series of NPP-VIIRS NTL data, we replaced it with NPP-VIIRS-like NTL data. Some previous studies have shown that the quality of NPP-VIIRS-like NTL data is similar to that of NPP-VIIRS NTL data, and can clearly reflect the details of the city and its changes in time series [69]. As shown in Table 5, NPP-VIIRS-like NTL presents a decreasing gradient in the region (urban areas, UFAs, and rural areas), consistent with the pattern of changes from urban areas to rural areas.

Remote sensing images can directly reflect the landscape characteristics of the region. In remote sensing images, the landscape characteristics of urban areas, UFAs, and rural areas are quite different, and can be readily distinguished. According to the economic development of each city in the Chengdu-Chongqing economic circle, we selected 6 representative regions (including two large cities, two medium-sized cities, and two small cities) for verification. Chengdu and Chongqing respectively contained 150 sample points; Dazhou and Leshan, respectively, contained 100 sample points; Guang'an and Ya'an, respectively, contained 50 sample points. The accuracy of the identification ranges from 72% to 77% (Table 6), and the average accuracy is 74.2%. After consideration, we believed there were several reasons for the poor results. First, UFAs are complex transition zones (e.g., land use, landscape, and spatial extent are dynamically changing). Second, the outer

side of UFAs is dominated by rural areas. During the verification process, this can lead to errors, i.e., considering UFAs as rural areas. Third, there are currently very few validation methods. Furthermore, differences in the development mode and natural environment of each city may reduce the accuracy. Due to these reasons, we considered the results to be acceptable. The sample points were randomly generated. Figure 17 shows the evaluation criteria for manual verification.

**Table 5.** Verification results of NPP-VIIRS-like NTL.

| City | Parameters | Urban Area (nW/cm$^2$/sr) | UFA (nW/cm$^2$/sr) | Rural Area (nW/cm$^2$/sr) |
|---|---|---|---|---|
| Chengdu | mean | 24.15 | 2.84 | 0.12 |
| | std | 17.27 | 4.47 | 0.64 |
| Chongqing | mean | 18.15 | 1.99 | 0.09 |
| | std | 12.07 | 3.49 | 0.80 |
| Leshan | mean | 10.77 | 1.53 | 0.03 |
| | std | 8.42 | 2.61 | 0.39 |
| Dazhou | mean | 22.69 | 1.54 | 0.03 |
| | std | 16.13 | 4.01 | 0.51 |
| Guang'an | mean | 18.84 | 2.45 | 0.09 |
| | std | 12.40 | 3.96 | 0.54 |
| Ya'an | mean | 17.13 | 2.33 | 0.09 |
| | std | 7.14 | 3.96 | 1.01 |

Note: the mean is the average NPP-VIIRS-like NTL in the region; and std is the standard deviation of NPP-VIIRS-like NTL in the region.

**Table 6.** Verification results of remote sensing images.

| City | Number of Sample Points | Accuracy |
|---|---|---|
| Chengdu | 150 | 74% |
| Chongqing | 150 | 77% |
| Leshan | 100 | 74% |
| Dazhou | 100 | 72% |
| Guang'an | 50 | 76% |
| Ya'an | 50 | 72% |

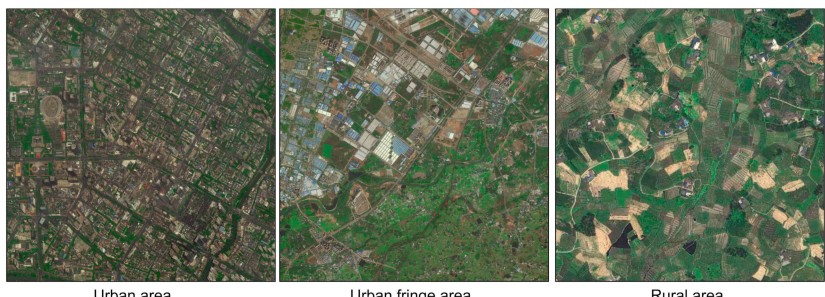

Urban area          Urban fringe area          Rural area

**Figure 17.** Evaluation criteria.

The two verification methods demonstrated that the identification results were broadly consistent with the observations (Figures A1 and A2). Compared with previous studies [23,28,32,35], remote sensing images were used to measure the accuracy of the identification results, and this approach can increase the credibility of the results. The method proposed to identify UFAs in this study can effectively reduce the influence of subjective factors, and can eliminate some noise points. The identification results can not only effectively reflect the spatial distribution, spatial morphology, and spatial structural characteristics of UFAs, but also can uncover the dominant factors controlling the expansion of UFAs. In addition, they can also provide a spatial extent reference for subsequent related studies.

*5.2. Interpretation of the Spatial Pattern of Educational Resources*

Past studies have mostly used administrative regions as a unit to analyze educational resources [70,71], but this approach is more macroscopic. Through the density-field-based hotspot detector and Geo-information Tupu, we explored the spatial pattern of educational resources at a finer spatial resolution. An optimized density-field-based hotspot detector can solve the problem of abnormal hotspots. This study enabled analysis of the spatial distribution characteristics and distribution patterns of hotspots, benefiting the realization of educational equity. There are still many shortcomings in this study. For example, the current method of determining the search radius has its advantages, but may not provide optimal results. In addition, further exploration of the time-series changes in the spatial pattern of educational resources and their influencing factors would be of great importance. It is also valuable to focus on the mechanisms of action for influencing factors (e.g., topography can influence the distribution of educational resources to control the formation of hotspots).

*5.3. Limitations and Future Study Directions*

The study has shortcomings which remain to be improved in the future; these are summarized as follows. First, build a standard identification and verification system for UFAs. There are many indicators that can reflect the characteristics of UFAs. In the future, we can consider incorporating population density, GDP, vegetation index, etc. Although remote sensing images can reflect the accuracy of the identification results, the accuracy can be affected by several factors (e.g., the complexity of the UFA itself). Second, explore the space-time evolution of UFAs. The direction, size, and speed of expansion can help us better understand urban development patterns. Third, mine the factors that influence the expansion of UFAs. Through geographical detectors and structural equation models, we can quantify the role of impact factors on the expansion of UFAs. Fourth, expand the scope of applications in UFAs. It is also important to evaluate the ecological sensitivity and the evolution of landscape patterns in the UFA. Finally, conduct in-depth analysis of how factors (e.g., physical geography, economics, policy, etc.) affect the distribution of educational resources.

## 6. Conclusions

Based on the characteristics of frequent changes of built-up land density in UFAs, we used a combination of urban boundaries, land use, spatially continuous wavelet transform, kernel density estimation, and reclassification, to identify the UFA of each city in the Chengdu-Chongqing economic circle in 2020. NPP-VIIRS-like NTL and remote sensing images were adopted to verify the accuracy of the identification results. Although the accuracy of identification results (74.2%) is not ideal, the methods used to identify UFAs and explore the spatial pattern of educational resources can be applied to other countries. Meanwhile, the method proposed in this study can effectively reduce noise points and the influence of subjective factors. The results reveal the characteristics of spatial morphology, spatial distribution, and spatial structure in UFAs. Based on the identification results, we explored the spatial pattern of educational resources by employing the density-field-based hotspot detector and Geo-information Tupu. The results highlighted differences in educational resources between regions. Policy, economic level, natural environment, etc., may be the main reasons for the differences.

Through the identification of UFAs and the analysis of educational resources, this study makes the following recommendations for urban planning. Scientific control is needed in fast-growing regions to avoid additional negative impacts. More investment is needed in slow-growing regions to promote balanced development. Areas with stable development can act as bridges to strengthen inter-regional linkages for coordinated development. The spatial distribution of educational resources can be optimized through policy guidance and improved transportation.

**Author Contributions:** Conceptualization, W.L.; Formal analysis, W.L.; Methodology, W.L. and Y.L.; Validation, W.L. and Y.W.; Visualization, W.L. and R.Z.; Writing—original draft, W.L.; Writing—review and editing, W.L., Y.L., R.Z. and Y.W. All authors have read and agreed to the published version of the manuscript.

**Funding:** This research was funded by the Fundamental Research Funds for the Central Universities, grant number SWU021003 and the National Natural Science Foundation of China, grant number 41571419.

**Data Availability Statement:** Not applicable.

**Conflicts of Interest:** The authors declare no conflict of interest.

### Appendix A

Figures A1 and A2 show the remote sensing images of some sampling points in the six cities.

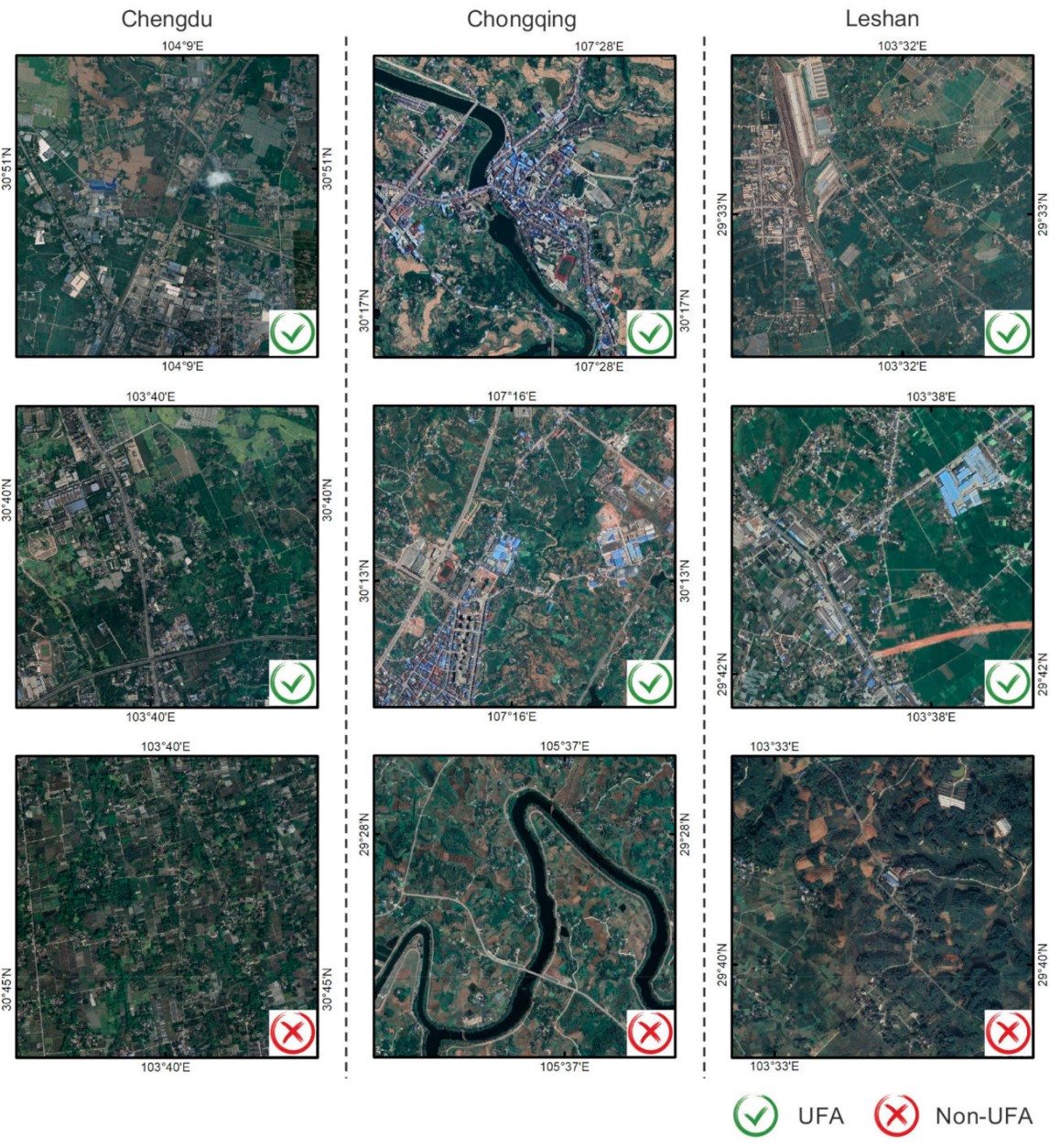

**Figure A1.** Remote sensing images of some sampling points in Chengdu, Chongqing, and Leshan. The coordinate reference system is the WGS-84 coordinate system.

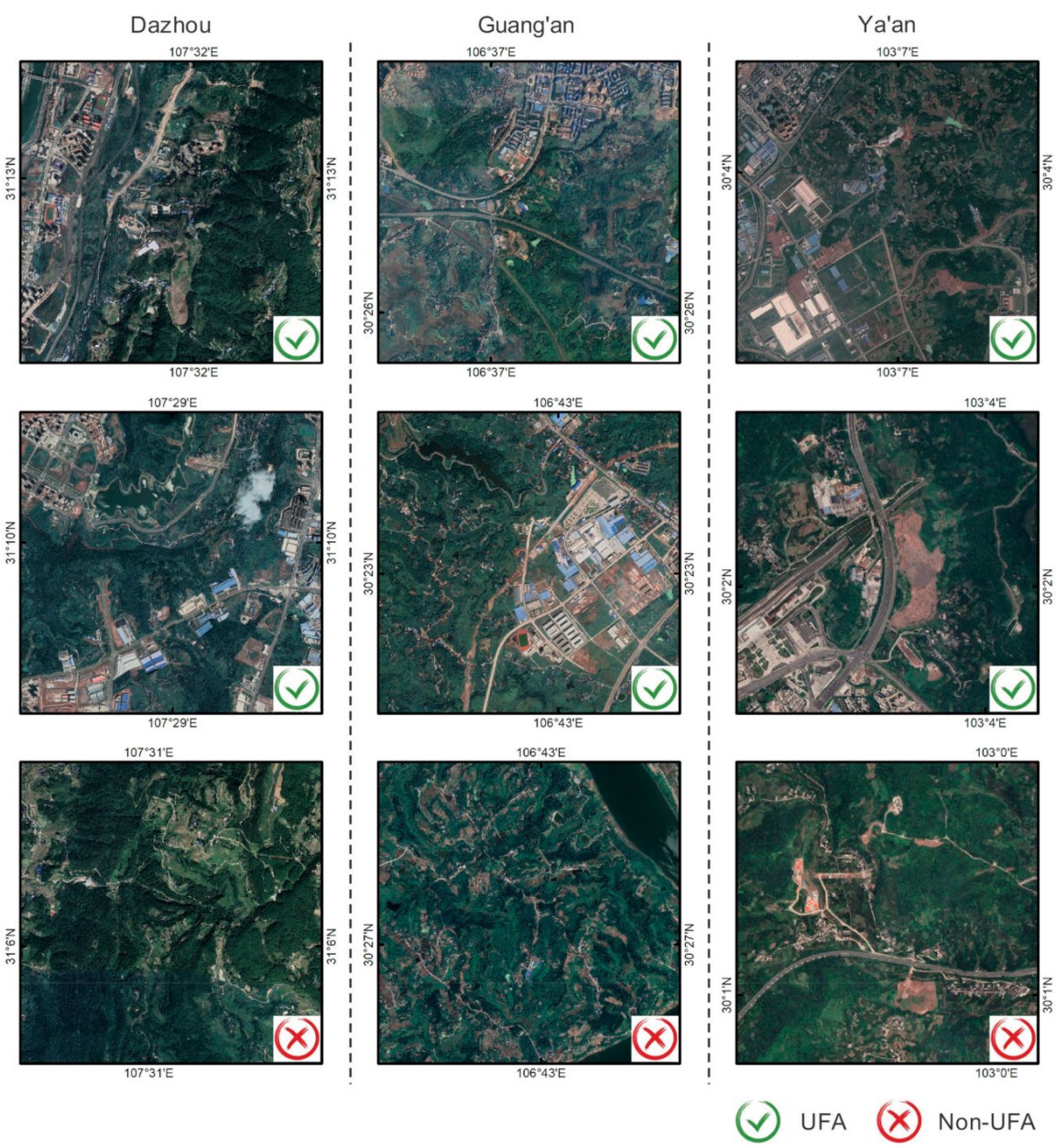

**Figure A2.** Remote sensing images of some sampling points in Dazhou, Guang'an, and Ya'an. The coordinate reference system is the WGS-84 coordinate system.

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
