# Peer review of "Using Remote Sensing to Identify Urban Fringe Areas and Their Spatial Pattern of Educational Resources: A Case Study of the Chengdu-Chongqing Economic Circle"

_remotesensing, doi:10.3390/rs14133148_

Round 1
Reviewer 1 Report
Dear Authors!
Thank you for the revised version of your manuscript, it was a pleasure to read it with the changes that you made.
However, I would like note that the manuscript was uploaded with the notes, changes and "Track Changes" and "All markup" enabled. After you correct this, I support the publication of the manuscript in this form.
Reviewer 2 Report
Line 36. UFA is singular, so frontier should be singular. Correct version: "As the frontier of urbanization, a UFA . . ."
Line 36. "A" or "an"? Use "a" since UFA starts with a consonant sound of "y." Correct version: "a UFA"
Lines 52 to 72. The parenthetical listing looks odd and each item is an incomplete sentence. I don't know if it's grammatically correct when done this way. It's acceptable to do this: "These are the following steps: (1) this is step 1, (2) this is step 2, and (3) this is step 3." But having each bulletted item as a separate sentence looks odd. SUGGESTION: Spell out each item and write complete sentences. For example, "First, select characteristic indicators to identify UFAs. . . . Second, detect mutation points of the characteristic indicators. . . . Third, . . . Finally, verify the . . ."
Line 79. "Education needs to be taken seriously" is an assertion. How about this? "However, few studies have focused on socio-economic issues in UFAs, especially those related to education. Some scholars . . ."
Lines 123-124. Use past tense since it happened already.
Line 138. What is "Amap"? SUGGESTION: "from Amap, a <give description here>."
Line 141. Correct term is "coordinate reference system."
Line 452. SUGGESTION: "Third, very few methods currently exist to validate . . ."
Line 479. SUGGESTION: "has its advantages but may not provide optimal results."
Line 483. Please build out this sentence. It sounds incomplete.
Lines 487 to 497. Please see comment for Lines 52 to 72.
Line 539. Show the coordinate reference system either on the map or in the caption.
Line 541. Show the coordinate reference system either on the map or in the caption.
Author Response
Please see the attachment.

This manuscript is a resubmission of an earlier submission. The following is a list of the peer review reports and author responses from that submission.
Round 1
Reviewer 1 Report
Dear Authors!
My comments regarding the article are the following.
- Structure and article desing: It should be imporved by following the journal guidelines. It is quite difficult for reviewers to read italic text through the whole article. The images are of good quality. Section numbering and naming should be structured even in a manuscript. The structure of some sections and paragraphs should be reconstructed (these are marked in the attached material).
- Research design: the artice itself is well-built up, but I miss the more detailed presentation of the sample area.
- Methodology: it is quite difficult, but easy to interpret through the text. Workflows should be described more accurately to increase reproducibility.
- Conclusions and the discussion is well-presented and intepretable.
- Reference style is not correct.
- English language is good.
Please see the attached file for more specific comments.

Reviewer 2 Report
- Formatting
- It seems the text format had reversed italic with non-italic text.
- Body of text should not be in italics.
- Section numbers not incremented.
- In terms of the English, this paper was very well written and easy to read. However, I noticed at least two instances of using a comma with a conjunction (and, or, ...) in non-compound sentences. Lines 159 and 297, possibly more. It's minor but noticeable.
- Content
-
Lines 317-319: Can the UFA rate metrics (<10%, 10%-30%, >30%) be specific to China or countries with relatively high growth rates? Would those same metrics apply to other global regions, say Europe or the United States? I think China is implied, but it may help to make this idea more explicit. Additionally, is there a quantitative basis for these values?
-
Lines 332-335: This paragraph makes certain assertions, one of the pitfalls of using passive verbs. Are these based on your study alone or other works? One of the goals of your study is to reduce subjectivity in determining UFAs, so it would help to back up these assertions with other scholarly works if only by reference.
-
Figure 12. Since you provided geographic coordinates, it may help to show the coordinate reference system.
-
Lines 362-367, 453: Impact of physical geography on educational resources. You mentioned how physical geography can impact the distribution of educational resources earlier in the paper. I think this would be an excellent line of follow-on research that you could include in your discussion.
-
Line 402: You wrote, "Verification and evaluation of identification results." It may help to make this more explicit with something like "Verification and evaluation of UFA identification results" to differentiate it from the educational resources part of this study.
-
Line 447: "Scale" can be an ambiguous term. Perhaps "finer spatial resolution" or something like that. Consider Werner Kuhn's core concepts of spatial information.
-
Lines 454-464: Limitations and future study directions. Could this study apply to other countries or do you think it is specific to China? I think addressing this question, even acknowledging it, would go a long way to making this paper more relevant to international readers.
-
Reviewer 3 Report
Regarding the article, it should be noted that the authors did not plagiarize: the publication is definitely worth a look. I think there is enough statistical material to process. Although the algorithm proposed by the authors is not new. However, the article is undoubtedly relevant. However, the analysis of the literature is not critical, but abstract. This does not provide reinforcement to prove the relevance of the topic.
Reviewer 4 Report
This manuscript tried to identify urban fringe area. Relationship between fringe and educational resources was analyzed. I believe this work is of some practical significance for urban planning. However, some concerns must be solved before it is acceptable for publication. Here I listed some comments for authors' references as follows:
- First of all, I am very sorry that the authors didn’t follow the basic formatting requirement of MDPI journal. It is quite difficult and annoying to read the whole manuscript with italic text. The numbering of reminder header is even completely wrong! In fact, from my point of view, it is fundamental and first-of-all that check the journal website for their formatting template from very beginning of submission. In this sense, the authors must re-format the manuscript to uniform the font size and font type to make the manuscript much easier to read!
- As a research article, the research gap is not well-introduced and analyzed in current version. I agree the scientific importance for UFA extraction, but what is the limitation, with in-deep analysis? Is it due to the lack of data source? Inadequate indicators considered? Or method/algorithm used? This is the basis of this work and the authors may clarify it with more detailed discussion in introduction section.
- Furthermore, here is my first biggest concern, why educational sources are analyzed? Obviously the authors didn’t mention much in the introduction. What is Tupu and how did so-called tupu apply to address the objective of this work?
- It is quite classical way used in this manuscript to derive the UFAs from multi-source data, which is okay. However, I didn’t see any detailed introduction that how data sources in Table 1 used for fringe extraction. At least the authors should give a brief workflow and current version is hard to follow.
- In fact, this work abstract the fringe area extraction to the mutation point identification. Mutation was detected by wavelet transform. However, how to set the center point to draw the profile line? This is quite important since if you select the wrong center point for your profile line (Fig. 4), it will lead to a failure for the whole detection process.
- Until now, I still can’t understand why education was analyzed, since it is quite strange to not linked to the main body of the methodology.
From the above-mentioned concerns, both overall quality and scientific novelty of this article requires significant improvement. A well-designed re-organization of the whole structure of the manuscript making it more compact and specific to some core issues in methodology should be done.
